# Drinking Water Supply in Rural Africa Based on a Mini-Grid Energy System—A Socio-Economic Case Study for Rural Development

**Joern Falk [1,\*], Björn Globisch [1,2], Martin Angelmahr [1] , Wolfgang Schade [1,3] and Heike Schenk-Mathes [4]**

[1] Fraunhofer Heinrich Hertz Institute HHI, Einsteinufer 37, 10587 Berlin, Germany;
bjoern.globisch@hhi.fraunhofer.de (B.G.); martin.angelmahr@hhi.fraunhofer.de (M.A.);
wolfgang.schade@hhi.fraunhofer.de (W.S.)

[2] Faculty II-Mathematics and Natural Sciences, The Technical University of Berlin, Straße des 17, Juni 135, 10623 Berlin, Germany

[3] IEPT, Clausthal University of Technology, Am Stollen 19A, 38678 Clausthal-Zellerfeld, Germany

[4] Management and Environmental Economics, Clausthal University of Technology, Julius-Albert-Straße 2, 38678 Clausthal-Zellerfeld, Germany; heike.schenk-mathes@tu-clausthal.de

\* Correspondence: joern.falk@hhi.fraunhofer.de

**Abstract:** Water is an essential resource required for various human activities such as drinking, cooking, growing food, and personal hygiene. As a key infrastructure of public services, access to clean and safe drinking water is an essential factor for local socio-economic development. Despite various national and international efforts, water supply is often not guaranteed, especially in rural areas of Africa. Although many water resources are theoretically available in these areas, bodies of water are often contaminated with dangerous pathogens and pollutants. As a result, people, often women and children, have to travel long distances to collect water from taps and are exposed to dangers such as physical violence and accidents on their way. In this article, we present a socio-economic case study for rural development. We describe a drinking water treatment plant with an annual capacity of 10,950 $m^3$ on Kibumba Island in Lake Victoria (Tanzania). The plant is operated by a photovoltaic mini-grid system with second-life lithium-ion battery storage. We describe the planning, the installation, and the start of operation of the water treatment system. In addition, we estimate the water prices achievable with the proposed system and compare it to existing sources of drinking water on Kibumba Island. Assuming a useful life of 15 years, the installed drinking water system is cost-neutral for the community at a cost price of 0.70 EUR/$m^3$, 22% less than any other source of clean water on Kibumba Island. Access to safe and clean drinking water is a major step forward for the local population. We investigate the socio-economic added value using social and economic key indicators like health, education, and income. Hence, this approach may serve as a role model for community-owned drinking water systems in sub-Saharan Africa.

**Keywords:** circular economy; drinking water; renewable energy; second-life battery storage; socio-economic development; sub-Saharan Africa (SSA); water safety in rural areas

## 1. Introduction

Access to clean drinking water and basic sanitation is not only indispensable for human health but a basic need and human right [1]. About 10% of all global diseases might be prevented with an improved water supply and associated hygiene facilities. For example, drinking water contaminated with faeces may transmit diseases such as diarrhoea, cholera, dysentery, typhoid, and polio [2]. Today, approximately 2 billion people get their drinking water from contaminated sources, which leads to approx. 485,000 deaths worldwide caused by diarrhoea only [3]. Thus, universal, safe, and clean access to drinking water contributes to the reduction of diseases and deaths, especially among children in

developing countries [4]. In the last 20 years, 1.6 billion people who did not have access to a basic drinking water supply got access to an improved source no more than 30 min away (round trip). Within the same period, the number of countries, in which less than 50% of the total population has access to drinking water has decreased from 25 in 2000 to only 11 in 2020 [5]. However, despite joint efforts by international actors, approx. 785 million people worldwide still do not have access to an improved source of drinking water. In addition, around 144 million of these people get their drinking water from untreated surface water sources such as rivers and lakes [6]. Rural areas are significantly less developed compared to urban regions and large differences exist between countries. As shown in Figure 1, the industrialised countries have succeeded in providing universal access to drinking water. In developing countries, however, especially in countries considered as the least developed by the United Nations, the level of supply is very low. Figure 1 shows that countries, where the population has only limited access to basic drinking water, are primarily located in the sub-Saharan region (SSA).

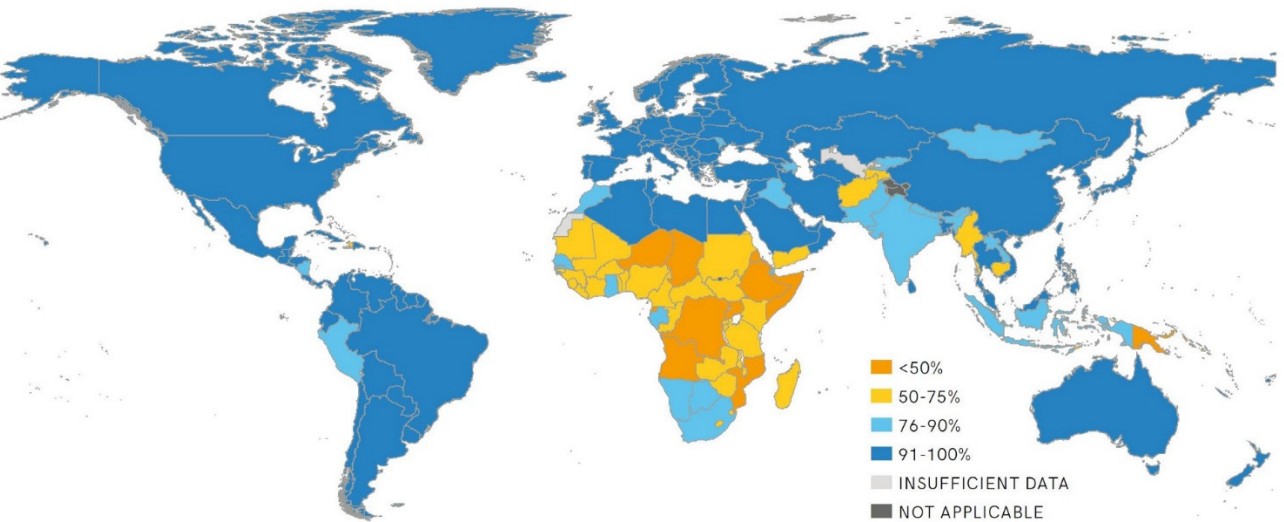

**Figure 1.** Proportion of population using basic drinking water services, 2017 (%) Source: WHO/UNICEF JMP (2019) Progress on household drinking water, sanitation, and hygiene 2000–2017. Special focus on inequalities.

This severe water shortage in some African countries might lead to the naïve assumption that the continent is fundamentally affected by a water shortage. However, Africa has considerable water resources. A large network of rivers and lakes are located on the continent, including the largest rivers such as the Nile, Congo, Niger, Zambesi, and Lake Victoria, which is the world's second-largest lake in terms of surface area. The volume of water stored in 677 freshwater lakes is around 0.03 million km$^3$. In addition, the continent has an immense amount of groundwater, estimated at 0.66 million km$^3$ [7]. Despite these abundant resources, the supply of clean drinking water is problematic. This is because either the resource is not evenly distributed or the water is polluted by industrial, agricultural, and domestic waste [8,9]. In view of the expected population growth in Africa, pollution will even increase in the future, as the degree of pollution correlates with the population growth [10]. The supply deficits can be attributed to the fact that the infrastructure programs of large players, such as the World Bank or regional development banks, focus on conventional measures. These include the so-called "first mile" infrastructure (central drinking water and wastewater systems coupled with private house connections). Usually, disadvantaged areas are not included in these programs as high per-capita investments would be required and financing opportunities to include disadvantaged population groups in this "last mile" supply chain are mostly lacking in many countries. Hence, alternative approaches for connecting disadvantaged areas have to be investigated [11].

Many interesting approaches have been developed in recent years. These include methods for low-cost water purification, reducing the use of chemicals, and reusing wastewater. However, also efficient desalination and technologies improve the decontamination of water [12]. The focus was on robust and autonomous technologies for water treatment systems [13]. Electricity is not always necessary for the treatment of drinking water, but it is very advantageous. Nevertheless, several solutions, which do not make use of basic electricity, have been investigated in the last years [14]. However, the success of such projects always depends on a sustainable economic concept that is adapted to the respective social context [15].

In this article, we present and discuss a case study on the installation of a drinking water filter system on the island of Kibumba in Lake Victoria (Tanzania, Africa). This system takes advantage of the basic availability of electricity offered by a solar-powered mini-grid system using second-life lithium-ion batteries as energy storage, which was installed on Kibumba in 2019 [16]. The central goal of this article is to highlight the socio-economic development perspectives in connection with the installation and commissioning of a drinking water filter system. In addition, the resulting possibilities to positively influence local hygienic conditions and consequently, the quality of life, are emphasized.

The article is organized as follows: in Section 2 we discuss the importance of access to both electricity and clean drinking water in more detail. In the latter case, we present the added value in a social, health, educational, and economic context. Subsequently, the initial situation of Kibumba Island as well as the conceptual scope of the installation of the drinking water system are described in Section 3. We explain the technical planning, installation, and commissioning of the system and how a supply of clean and safe drinking water was achieved. In Section 4, a self-managed financing model at cost price is presented. Different models for sustainable economic operation and expected long-term influences on the living conditions of the residents are discussed. In analogy to Section 2, we analyse socio-economic added value for the local people in a social, health, educational, and economic context in Section 5 before we summarise our results in Section 6.

## 2. Electricity and Drinking Water Supply as Basis for Rural Development

### 2.1. Electricity in Rural Areas

An essential cornerstone of socio-economic development is a well-functioning infrastructure. Over the past ten years, national and international actors have made great efforts to improve important components of the public infrastructure. In particular, the rural regions of Africa, south of the Sahara, were the focus of development activities, especially to promote electrification. Despite various programs and collaborative efforts of different parties, the provision of electricity outside urban habitats in SSA remains inadequate. In this region, an estimated 600 million people are still cut off from a comprehensive and reliable energy supply [17]. To supply these areas, small, off-grid solutions, so-called mini-grid systems, have been established in recent years. Depending on country-specific resources, these systems can use different technologies, such as wind power, photovoltaics, hydropower, geothermal energy, and biogas, to provide electricity flexibly and cost-effectively [18]. Even if there are technical solutions for drinking water supply without direct electricity demand [19], an unrestricted power supply throughout the day is of great advantage for elementary components of this infrastructure Mini-grid systems, generate electricity from a wide range of renewable energy sources, have the disadvantage of not being able to ensure this without suitable storage systems as they use resources, which are only temporally available (wind, sun, hydropower). Often, these gaps in supply are compensated for with aggregates that generate energy by burning fossil fuels. However, this has the major disadvantage of producing a high level of harmful emissions on site. In addition, these systems are sometimes unreliable and expensive to operate and maintain [20]. Therefore, battery storage integrated into the system, especially lithium-ion batteries (LIBs), are suitable for compensating for these technology-related supply gaps [21]. LIBs offers an ideal basis due to their high energy density, high performance, and long service life. Since LIBs are

already used extensively in the automotive industry the volume-weighted market price per kWh has fallen by 89% to 137 USD in the years from 2010–2020 [22]. Depending on the manufacturer, LIBs in electric vehicle applications consume only 20–30% of their total capacity output, which opens an interesting market for so-called second-life batteries (SLBs). These also include former traction batteries used as storage systems [23] with a price of 51 USD per kWh [24]. With the push for electromobility, this development will continue to increase in the coming years and is of both economic and ecological relevance [25]. On Kibumba Island, an SLB has been in use as an electricity storage facility since 2019, which will be used for the continuous operation of the drinking water filter system presented in this article.

*2.2. Access to Clean Drinking Water*

Access to clean and safe drinking water is an essential factor in assessing living conditions. Social and economic factors are significantly influenced by it, which in turn has a direct impact on people's life expectancy. Accordingly, socio-economic development is inextricably linked to this [26]. As shown in Figure 1 developing countries, in contrast to industrialised countries, have a conspicuous deficit in access to clean drinking water and sanitation facilities [27]. According to a 2019 UNICEF report, 1.4 billion people had access to a basic service, which is characterized by better sources than a limited service and can be reached within a 30-min walking radius (round trip). Furthermore, 206 million people used limited service, which is outside the 30-min radius [28]. In addition, there are differences between rural and urban populations as well as between high-income and low-income groups in these countries. The resulting effects have a direct influence on socio-economic development [29]. This imbalance is particularly pronounced in SSA and is a major obstacle to successful development [30]. To improve their situation, low-income earners are dependent on measures initiated by their government or other actors, which are sometimes implemented irregularly and with a long delay. In contrast, people with higher income can improve their situation autonomously, for example by self-supply through self-drilled wells or the use of local premium sanitation services. The associated benefits: increased hygiene, increased productivity, reduced susceptibility to disease, etc., lead to a significantly increased quality of life, which directly influences socio-economic development in a positive way [31]. In the following subsection, we describe this effect in more detail and analyse how the sub-items social context, health, education, economy, and income are all intrinsically linked. The summary of this analysis is depicted in Table 1, which displays the spectrum and the impact of the socio-economic added value through access to clean and safe drinking water categorized by the respective sub-item.

**Table 1.** Summarised socio-economic added value through access to clean and safe drinking water.

| Indicator | Spectrum | Impact |
|---|---|---|
| SOCIAL CONTEXT | Social status<br>Time savings | Respect by neighbouring communities<br>Stronger community cohesion<br>More time for community activities |
| HEALTH | Lower health risks due to diseases<br>Lower health risks from assaults | Improved quality of life<br>Increased life expectancy |
| EDUCATION | Less school absences | More time for learning<br>Better school graduation<br>Less risk of poverty |
| ECONOMY/INCOME | Less days of illness<br>Less treatment costs | Increased labour productivity<br>Increased wealth |

2.2.1. Social Context

The limited availability of clean drinking water has serious consequences. It is particularly problematic for households with very low incomes to cover their daily needs, as they usually lack the financial means to switch to alternatives such as bottled water. This puts

additional pressure on household incomes, which are already heavily burdened [32]. In addition, access to clean and safe water sources and sanitation has a multi-faceted impact on the psyche of communities. Installed water systems are perceived as a status symbol and strengthen the community and its cohesion. Compared to households or communities without such access, this creates additional social pressure and a social divide. In addition, women and children are predominantly responsible for the procurement of water, which results in either a burden or a relief for the weakest members of the community [33]. The elimination of the time-consuming process of fetching water results in considerable time advantages for these groups of people. This can be used to pursue other activities, such as social activities for the community, gainful employment, or educational activities. Another significant point is that the sense of security is strengthened for these groups of people, as violence and assaults are not uncommon while fetching water at distant and remote water points [34].

### 2.2.2. Health

Clean and safe drinking water is essential for human nutrition and hygiene. World-wide, approximately 3,575,000 people die each year from diseases that can be traced back to contaminated or polluted water, the majority (2.2 million) being children [4].

In rural areas without significant industry and agriculture, human waste is the main cause of polluted water. Hence, part of the problem could be solved by dealing with it responsibly and protecting natural sources [35]. Studies show that water bodies are contaminated with harmful viruses and bacteria, such as hepatitis A and C viruses, polyomaviruses, or human adenoviruses and enteroviruses [36]. Furthermore, there is additional exposure to heavy metals, nitrates, fluorides, and other chemicals that can be traced back to industrial and agricultural influences. In particular, the extraction of raw materials or the refining of raw oil contributes significantly to the pollution of water resources.

In households without adequate supplies, the quality of life of those affected is therefore usually comparatively much worse. This is because contaminated water and associated diseases are among the main causes of high child mortality in many developing countries, with young children being particularly affected [37].

According to this, the health of people is directly influenced positively by the improvement of water supply and sanitation facilities [27]. In particular, this is achieved through the associated reduction of diarrhoeal diseases and other related health risks. Since water is often particularly contaminated by coliform germs, faecal bacteria that pose a health risk. These pathogens attack the immune system and trigger diarrhoeal diseases and urinary tract infections, for example [38]. Clean drinking water can also significantly reduce the risk of infection with diseases such as schistosomiasis, hepatitis A and the highly contagious eye infection trachoma. In summary, universal access to safe drinking water could prevent more than 390 million cases of diarrhoeal diseases and their health consequences worldwide each year [39].

### 2.2.3. Education

In sub-Saharan African countries, about 75% of households meet their water needs from a source outside the home. The main responsibility for fetching water is borne by women and children in 71% of households, with the average time spent doing so being about 30 min each way. Depending on the number of people living in the household, water may have to be fetched several times a day. Since the water is mostly obtained from unclean sources, it has to be boiled for basic disinfection. The firewood needed for this is also mainly procured by children. The average time required for this is 10 to 65 min per day [40]. Direct access to clean drinking water can significantly reduce these time expenditures as well as contribute to relieving the burden on groups of people. Positive impacts can be felt especially on children and their individual education. The time saved reduces absences from school, creates additional opportunities for other social activities, and increases the time available for learning and education. In addition, the consumption of clean drinking

water significantly reduces the risk of falling ill, thereby reducing absenteeism due to illness. Increased participation in schooling leads to higher school completion rates and is an essential factor for further local development [41]. This is a key factor in reducing poverty, as the risk of poverty falls by around 11% with the completion of elementary school. With the completion of secondary school, this value increases to as much as 43% [42]. It is, therefore, to be expected that with a higher level of education, more income can also be earned in the future.

### 2.2.4. Economy and Income

Proximity to a safe and clean water source correlates with comparatively increased wealth [43]. Consequently, a lack of clean water can be a major obstacle to economic development, because not only the people themselves depend on it, but also many economic sectors. Considerations in this regard are mainly focused on factors such as time savings and health aspects as well as their monetary effects. For this purpose, incomes are calculated that could be earned during the time in which water would normally have to be procured. However, times in which one cannot be gainfully employed due to the consumption of polluted water, including possible treatment costs, are taken into account [44]. A reduction in sick leave also has a positive impact on labour productivity in the long term. Studies by the WHO have shown that access to clean drinking water and sanitation can reduce illness-related absenteeism by 10%. This corresponds to a global macroeconomic added value of 84 billion USD [45].

### 3. Drinking Water Supply and Electricity on Kibumba

The island of Kibumba on Lake Victoria belongs to the Mazinga Village Council, a group of islands with a total of 1600 inhabitants in northwest Tanzania. On Kibumba there is a primary school and a health post, each serving a catchment area of about 8500 inhabitants of the overarching Mazinga Ward. Despite the island's immediate location in a freshwater lake (Figure 2), the supply of clean drinking water is a major problem for the local people. Well drilling for drinking water supply is not possible in this area because Lake Victoria lies on the central plateau, which is characterised by tertiary gneisses and granites [46]. Therefore, the only source of clean drinking water is mineral water in plastic bottles, which has to be transported from the mainland to the island and is a financial burden on households. As the recycling of plastic bottles is practically unknown, large environmental impacts are caused by plastic waste or by its incineration. The people on Kibumba cover their daily water supply mainly with water that is directly taken from Lake Victoria. However, this water is heavily contaminated with viruses, bacteria, and schistosomes [47]. The latter parasite is the main cause of the infectious disease schistosomiasis, one of the most widespread parasitic infectious diseases along with malaria. Every year, about 200,000 people die as a result of a schistosomiasis infection [48]. Humans are infected when larval forms of the parasite enter the body via the skin. According to WHO estimates, about 230 million people worldwide are infected with schistosomiasis, most of them in Africa. Tanzania has the second-highest infection rate with about 19 million people [49].

Hence, access to clean water is a severe challenge for people living on Kibumba. In the following subsection, the initial situation on Kibumba Island, the technical planning and design as well as the installation and commissioning of the drinking water system are outlined. The socio-economic added value that can be expected from the installation of the system is also presented in detail.

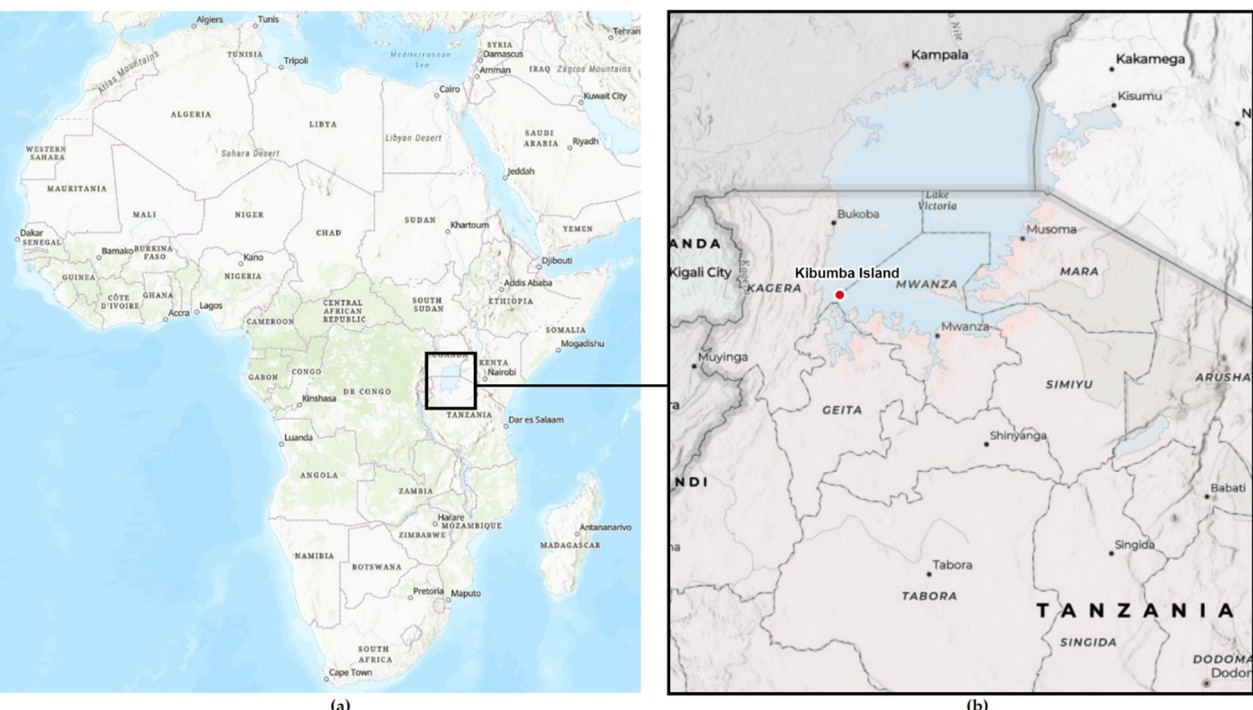

**Figure 2.** Location of the study (**a**) Map of Africa marked with a black box showing the location of Tanzania (**b**) Kibumba Island, Lake Victoria, showing location in Tanzania with a red dot (Source: Esri Arcgis, HERE, USGS, NGA, 2022).

### 3.1. Planning and Design

The drinking water filter system takes advantage of a solar-powered mini-grid system with second-life lithium-ion batteries as energy storage. The system was already installed in 2019 [21]. Figure 3 shows a schematic of the complete system. It consists of a solar-based energy generation system with battery storage to provide energy even during the night hours. Due to the regionally high solar radiation of 2000–2500 kWh/m$^2$/year [50], this plant can provide around 15,000 kWh annually [16]. The system thus forms an optimal basis for the expansion of public services as well as the operation of a drinking water system.

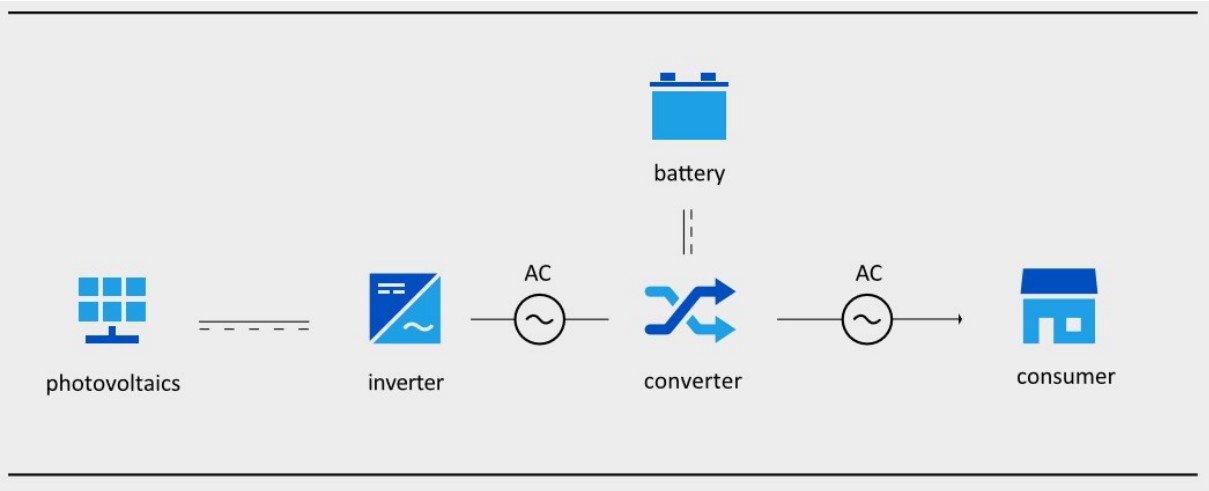

**Figure 3.** Schematical Setup of the Mini-Grid System on Kibumba Island, Tanzania.

On Kibumba, the first provisional water pipeline was laid in 2020. Here, water was directly taken from Lake Victoria with an electric pump and channelled to the local fishery.

The fishermen used the water to clean their catch before freezing it. This provisional water supply was expanded into a drinking water supply according to the WHO standard in 2022. This standard specifies the minimum requirements for safe drinking water and is visualised in detail in Table 2.

**Table 2.** WHO standards for drinking water quality based on WHO guidelines for drinking water quality (WHO 2017).

|  | **Microbial Aspect** | **Chemical Aspect** | **Radiological Aspect** | **Acceptability Aspect** |
| --- | --- | --- | --- | --- |
| Detailed aspects | Bacteria<br>Viruses<br>Helminths<br>Protozoa | Elemental chemicals<br>Chemical compounds | Radioactive elements | Neutral taste<br>Neutral odour<br>Clear appearance |
| Acceptance criteria | Undetectable<br>0 cells/mL | Limit values<br>per mg/L | Range between<br>10–10,000 Bq/L | No foul or bitter taste and smell<br>No dirty look |

The average water consumption in Tanzania is about 25 L per person per day [51]. This is very low compared to other countries such as Germany with 123 L [52] and the USA with 560 L [53] per person per day. In order to supply both the 300 people living on Kibumba and the immediate vicinity with sufficient drinking water, the plant was designed with a treatment capacity of up to 30,000 L per day. In preliminary discussions with the responsible community representatives, the average daily water demand of initially around 9000 L was determined. Hence, a capacity of 30,000 L is able to fulfil the current need and can absorb a future increase in consumption as well. A system with a 5000- and a 500-L water tank connected to appropriate mechanical filtration, chlorination, and UV disinfection was designed and later installed on site. Figure 4 shows a schematic setup of the water treatment process. The necessary filtration stages were determined by analysing water samples from local authorities. First, the water from the pipeline passes through a sand filter that removes suspended matter. This is followed by a chlorination stage that kills bacteria, viruses, and other pathogens, after which the water is fed into a 5000-L pre-treated water storage tank. Afterwards, the water is passed through a Hydra cartridge filter with 50 μm mesh size to remove sediments and three sequential filter cartridges with 25 μm, 10 μm, and 1 μm mesh size to filter all residual sediments and reduce microorganisms. After filtration, the water passes through granular activated carbon and afterwards through a carbon block. This reduces the chloride content and organic chemicals and results in a reduction of odour, improving the taste. In the final step, a UV purification unit and reverse osmosis filtration are used to kill any remaining bacteria and viruses, including legionella, salmonella, faecal coliforms, *Escherichia coli*, influenza, hepatitis, and dysentery, as well as dissolved solids. At the end of the supply line, drinking water can be taken by the local population. To ensure the perfect quality of the drinking water, the water is tested daily for chemical and microbial contamination, and no contamination has been detected. The costs of the water treatment setup are discussed in detail in Section 4.3.

*3.2. Assembly and Commissioning*

In In preparation for the installation of the system, a hut with a footprint of about 2 m$^2$ was built for the water pump and provided with a power connection, visualised in Figure 5a. Sturdy concrete bases and a stable table with a metal frame were made for the two water tanks and the water tap, as visualised in Figure 5b,c. The cable and earthworks took about two working days in advance. The technical equipment included components with a total weight of about 500 kg. Due to the local infrastructure, the transport goods had to be reloaded several times, most recently from the truck to the boat, see Figure 5d. Four working days were needed to install and commission the system, followed by an initial test phase. During the test measurements, an average throughput of 1000 L per hour was achieved. Extrapolated to the daily operating time, the calculated daily demand of 9000 L was achieved, Figure 6 shows the fully installed and operational system.

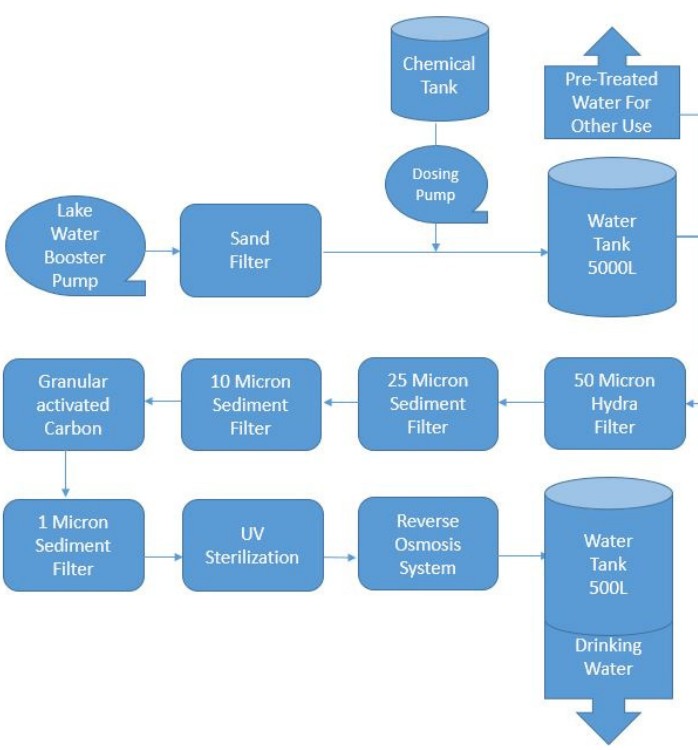

**Figure 4.** Schematic of the Water Treatment Procedure (based on Manufacturer's description).

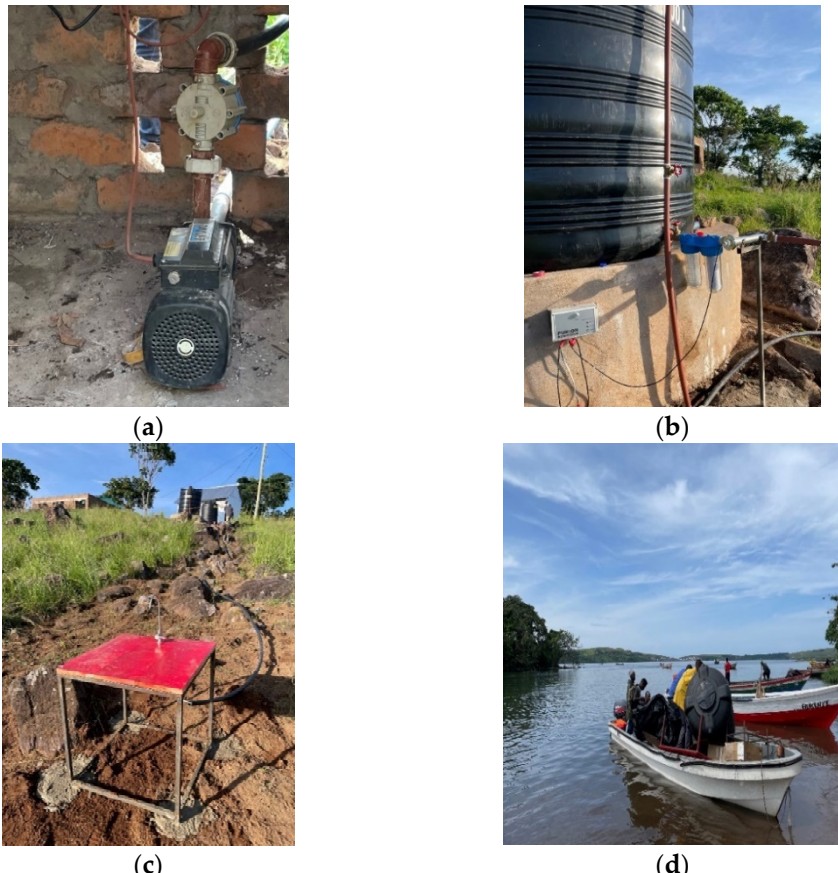

**Figure 5.** Pictures onsite (**a**) Booster pump, (**b**), Tank with filter system (**c**) Water tap, (**d**) Logistics.

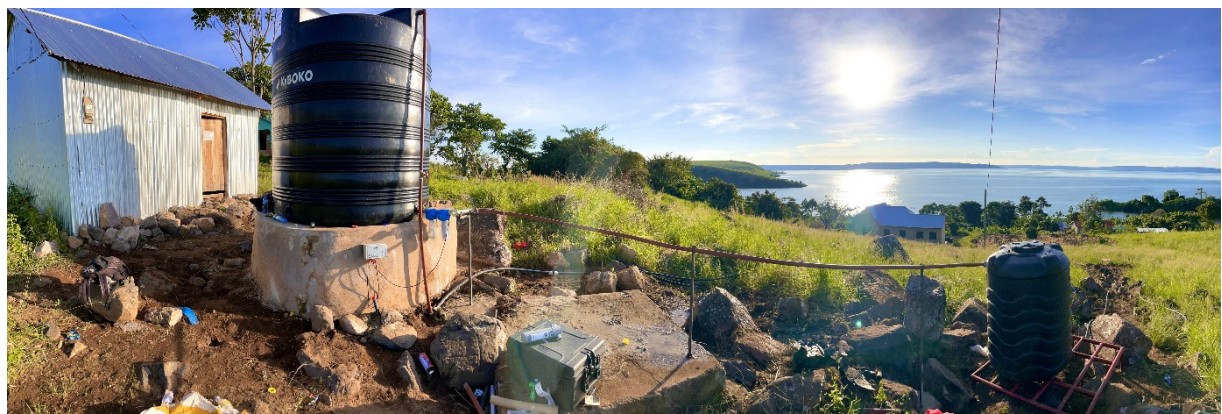

**Figure 6.** Picture of drinking water system on Kibumba.

## 4. Model for Sustainable Operation

Due to a large number of different projects and their individual complexity, it is very difficult to obtain valid data on the economic efficiency of drinking water facilities in African rural areas. This insufficient data makes it difficult to accurately compare economic evaluation criteria [15]. In order to obtain an evaluation basis for the scenario on Kibumba Island, average comparative prices for water offers directly from the region on the west coast of Lake Victoria were used. Samples were taken at nine locations between Mwanza and Nazinga. Based on these values, the respective average prices for bottled water, commercial tap, and (shared) so-called neighbours tap were determined and summarised in Table 3. The average water price at commercial taps was 0.00193€/L, at shared fixed water connections of private households, the price was 0.00096€/L, and commercial bottled drinking water in shops is the most expensive way to get drinking water, with an average price of €0.36/L. This price is also roughly in line with current international surveys of €0.385/L [54]. Hence, the newly installed drinking water supply system should be able to offer clean drinking water for a price that is competitive to all the existing solutions. In the next subsections, we will discuss different financial models for the operation of the system and amortization scenarios of the system. However, the costs for the electricity supply (system costs and electricity costs) are not taken into account here, as the electricity required for the drinking water system is provided free of charge by the national energy service provider JUMEME, with the support of public authorities, in order to promote development on Kibumba.

**Table 3.** Average Prices Commercial Drinking water west coast of Lake Victoria (From Mwanza to Kasharunga).

| Source | Common Unit in L | Price/L in TZS | Price/L in EUR [1] |
|---|---|---|---|
| Commercial tapping points | 20 | 5 | 0.00179 |
| Neighbour's tap (through public supplier) | 20 | 2.5 | 0.00089 |
| Shops (bottled) | 1 | 1000 | 0.36 |

[1] Based on the official exchange rate of Tanzanian Shilling (TZS) to EUR (1 EUR = 2797.29 TZS; German Bundesbank 2021).

### 4.1. Sustainable Financing by Individual Responsibility

Maintaining self-sufficient water systems in rural Tanzania is a complex challenge. Despite years of investment by the government and other stakeholders, about 50% of the population does not have access to safe drinking water. One major reason for this is that up to 20% of the newly created water points fail within the first year after their installation [55]. This was often due to the fact that actors from outside the community were responsible for the management. As a result, individual adaptations to the needs of the local people were usually not taken into account, which had a negative impact on the

usefulness and acceptance. Therefore, community-based, decentralised water management is now considered indispensable for the success of such projects [56]. The community's ownership of the operation and maintenance of the facility promotes careful and far-sighted handling. This approach is supported by The Water Supply And Sanitation Act [57] in which the state encourages Community Based Water Supply Organizations in planning, implementation, and operation realized by appropriate persons [58].

### 4.2. Conceptual Framework

For a sustainable economic operation of the plant, it is indispensable to involve all stakeholders such as islanders, community members, and political and economic actors in the development of a suitable financing model. This is the only way to ensure long-term acceptance and understanding [59]. The aim is to operate the facility at its cost price. To this end, the consumables necessary for unrestricted operation must be taken into account. Maintenance and servicing work must also be included. For an autonomous operation, local people have to learn the technical details of the system, such that necessary service and maintenance work can be largely realised by the corresponding persons on site. The sale of the drinking water produced by the plant is intended to generate income to finance the ongoing operation and to build reserves for necessary future investments in the plant. As the plant was provided as part of a community development project and no private investor was involved, no loan payments had to be considered. The duration of operation of the facility was assumed to be 15 years.

### 4.3. Calculation

The components that were used for the drinking water system installed on Kibumba are summarised in Table 4. This solution is tailored to Kibumba's needs and unit prices correspond to real market prices. The total equipment including logistics and installation costs is calculated at 29,791,139 Tanzanian Shilling (TZS), which corresponds to a total value of approximately 10,640 EUR (Based on the official exchange rate of Tanzanian Shilling (TZS) to EUR (1 EUR = 2797.29 TZS; German Bundesbank 2021)). The annual maintenance and operating costs required for operation as well as the financing costs are listed in detail in Table 5. These were determined individually for the Scenario on Kibumba by the regional supplier of the plant, who knows the local conditions precisely and can assess them empirically.

**Table 4.** Applied components for drinking water system.

| Item Description | Quantity | Unit Price in EUR |
|---|---|---|
| Submersible pump 1.1 kW | 1 | 221 |
| Booster pump, pedrollo CPm 170 0.75 kW | 1 | 160 |
| Pressure tank, pressure Switch, pressure gauge | 1 | 126 |
| Sand filter CX 600 | 1 | 676 |
| Water tank 5000 L | 1 | 800 |
| Water tank 500 L | 1 | 100 |
| UV purifier 25 W | 2 | 149 |
| Filter bodies 10″ | 4 | 13 |
| Cartridge filter 10″ | 3 | 7 |
| Acid washed carbon | 1 | 16 |
| Chemical tank 170 L with mixer | 1 | 854 |
| Dosing pump | 1 | 416 |
| Chlorine chemical 5 kg | 1 | 18 |
| Hydra self-cleaning filter | 1 | 70 |
| Reverse osmosis system 200 L | 1 | 3190 |
| Plumbing materials | 1 | 1493 |
| Electrical materials | 1 | 640 |
| Installation labour charge | 1 | 1240 |
| Transport charge | 1 | 249 |
| Total | 23 | 10,640 |

**Table 5.** Annual maintenance, operating, and financing costs (Manufacturer's specification).

| Annual Costs | | | | | | Price in EUR |
|---|---|---|---|---|---|---|
| Maintenance and Operating | | | | | | 1100 |
| General consumables | | | | | | *200* |
| General maintenance | | | | | | 200 |
| Spare parts | | | | | | 400 |
| Salaries (repair) | | | | | | 300 |
| Financing costs | | | | | | 1970 |
| Year | Debt Level Start of Year | Annual Rate | Interest (16.68%) | Repayment | Debt Level End of Year | |
| 1 | 10,640.00 | 1969.47 | 1774.75 | 194.72 | 10,445.28 | |
| 2 | 10,445.28 | 1969.47 | 1742.27 | 227.20 | 1218.09 | |
| 3 | 10,218.09 | 1969.47 | 1704.38 | 265.09 | 9953.00 | |
| 4 | 9953.00 | 1969.47 | 1660.16 | 309.31 | 9643.69 | |
| 5 | 9643.69 | 1969.47 | 1608.57 | 360.90 | 9282.79 | |
| 6 | 9282.79 | 1969.47 | 1548.37 | 421.10 | 8861.69 | |
| 7 | 8861.69 | 1969.47 | 1478.13 | 491.34 | 8370.35 | |
| 8 | 8370.35 | 1969.47 | 1396.17 | 573.29 | 7797.05 | |
| 9 | 7797.05 | 1969.47 | 1300.55 | 668.92 | 7128.13 | |
| 10 | 7128.13 | 1969.47 | 1188.97 | 780.50 | 6347.64 | |
| 11 | 6347.64 | 1969.47 | 1058.79 | 910.68 | 5436.96 | |
| 12 | 5436.96 | 1969.47 | 906.88 | 1062.58 | 4374.37 | |
| 13 | 4374.37 | 1969.47 | 729.65 | 1239.82 | 3134.55 | |
| 14 | 3134.55 | 1969.47 | 522.84 | 1446.63 | 1687.92 | |
| 15 | 1687.92 | 1969.47 | 281.55 | 1687.92 | 0.00 | |
| Total | 10,640.00 | 29,542.03 | 18,902.03 | 10,640.00 | 0.00 | |
| | | | Total | | | 3070 |

Table 6 summarizes the calculations. Column #1 represents the maximum annual drinking water production of the system, column #2 shows the actual current consumption, water price at cost is calculated in column #3, and revenues expected from the sale in column #4. The calculation of a representative drinking water price as well as the economic efficiency of the system depends on different variables. These are location, installation costs, and operating and maintenance costs.

**Table 6.** Assumptions calculations.

| Assumptions Calculations | |
|---|---|
| #1 | Annual drinking water production: <br> Annual production in $m^3$: Daily production in $m^3 \times 365$ days $= 30\ m^3 \times 365$ days $= \underline{10{,}950\ m^3}$ |
| #2 | Current annual consumption: <br> Annual consumption in $m^3$: Daily consumption in $m^3 \times 365$ day $= 12\ m^3 \times 365$ days $= \underline{4380\ m^3}$ |
| #3 | Water price at its cost: <br> Price per $m^3$ in EUR: $\dfrac{\text{Annual cost (operating+credit) in EUR/year}}{\text{Annual consumption } m^3/\text{year}} = \dfrac{(1100+1970)\ \text{EUR/year}}{4380\ m^3/\text{year}} \approx \underline{0.70\ \text{EUR}\ [1]/m^3}$ |
| #4 | Annual revenue: <br> (a) Cost price in EUR: <br> Annual consumption in $m^3$/year $\times$ price per $m^3$ at cost <br> $= 4380\ m^3$/year $\times 0.70\ \text{EUR}/m^3 = 3066\ \text{EUR/year}$ <br> (b) Market price at neighbourhood tapping points in EUR <br> Annual output in $m^3$/year $\times$ price per $m^3$ at market price at neighbourhood tapping points <br> $= 4380\ m^3$/year $\times 0.89\ \text{EUR}/m^3 = 3898\ \text{EUR/year}$ <br> (c) Market price commercial tapping points in EUR: <br> Annual output in $m^3$/year $\times$ price per $m^3$ at market price at commercial tapping points <br> $= 4380\ m^3$/year $\times 1.79\ \text{EUR}/m^3 = \underline{7840\ \text{EUR/year}}$ |

[1] Rounded to 2 decimal places.

Even though in this case the investment costs of 10,640 EUR for the plant were financed by third parties, Tables 5 and 6 show a hypothetical calculation of potential financing costs. The calculation in Table 6 Column #4 (a) shows that the revenue to be generated from the plant at cost price is high enough to almost completely cover the annual operating and maintenance costs as well as potential loan costs of 16.68% [60] during the expected lifetime of 15 years. This does not take into account the projected average inflation rate of 4.4% [61]. Thus, the annual maintenance costs would almost double during this period with a value of 2010 EUR. In this context, however, the comparative prices (Neighbour's Tap and Commercial Tap) would also become correspondingly more expensive. In this presentation, we are initially only concerned with the hypothetical competitive character of the system. A corresponding database for economic parameters such as inflation and wage development should be recorded and evaluated in detail in further research work.

As shown in Table 6, ASSUMPTIONS CALCULATIONS column #1, the capacity of the plant is 30 m$^3$ per day, which corresponds to a total annual production of 10,950 m$^3$/year. The current annual consumption of 4380 m$^3$ leads to a cost price of 0.70 EUR/m$^3$, which includes the annual operating and financing costs. Currently, according to Table 6, column #4 (a), annual revenues of EUR 3066 can be expected. This would make the plant at cost price almost cost-neutral for the operating community over the entire useful life of 15 years. Under calculation column #4 (b), the comparison is made to the market price at neighbourhood connections, which is 0.89 EUR/m$^3$. Calculation column #4 (c) shows the same based on the average market price of 1.79 EUR/m$^3$ for clean drinking water on the west coast of Lake Victoria, resulting in annual revenues of 7840 EUR/year. Comparisons (b) and (c) only serve to additionally classify the drinking water system under other existing market conditions.

As indicated in Table 3, there are three options available for the safe and clean supply of drinking water on Kibumba, bottled water, water from commercial taps and water from neighbourhood taps. However, all of these can only be sourced from the mainland which leads to additional effort and cost. The most expensive is bottled water in shops, which costs 360 EUR/m$^3$.

At commercial tapping points, costs correspond to 1.79 EUR/m$^3$, while neighbourhood tapping points cost only half that at 0.89 EUR/m$^3$.

### 4.4. Payment Options

In order to offer the most suitable payment option for local people, it is important to take their current living conditions into account. This does not only include the average household income but also how people earn their money. This is because the decision for a possible payment option depends significantly on whether fixed salaries are received, or the income consists of irregular ad hoc employment. Discussions on site have shown that essentially no jobs with fixed salaries exist on Kibumba Island and thus people do not have constant household income. With a share of 60%, the majority of all households live from fishing, 35% earn their living as farmers, and 5% live from small businesses. The average daily income per household on Kibumba Island is about 6000 TZS, which corresponds to 2.14 EUR [62], according to individuals and officials. Hence, methods for flexible and demand-oriented payment need to be found. Basically, the cost price for the water produced by the system (see Table 6 #3) is a solid basis. The average household size in Kibumba is five persons. With a daily water consumption of 25 L per person, 125 L of clean and safe drinking water costs just about 0.09 EUR. This amount can be paid even if the household income is low.

### 4.4.1. Monthly Subscription

With this payment method, customers pay a regular (monthly, quarterly, half-yearly, yearly) constant fee for certain services. In the specific scenario described in this article, it would be conceivable to create fixed contracts based on the estimated consump-

tion per household. For the operator, the fixed turnover facilitates the calculation of maintenance costs.

### 4.4.2. Pay-As-You-Go

In recent years, the pay-as-you-go procedure has been successfully introduced in developing countries for a wide range of services [63]. It is a non-contractual service where customers initially pay only a one-off fee for general access. Subsequently, only the real consumption is charged. From the user's point of view, this offers some decisive advantages. The barrier to entry is low, as there is no fixed contract, which requires regular payments. In the case described here, it offers users the flexibility to access water on demand, without making any commitments. From the operator's point of view, however, this payment method is associated with certain risks. Calculating the expected turnover is difficult, as customers consume on-demand and fluctuations are usual. In addition, the permanent system costs have to be pre-financed by the operator.

### 4.4.3. Preferred Payment Option

Considering the economic situation of the local households and their irregular incomes, fixed monthly payments were considered unsuitable as they do not allow for a flexible response to the respective monthly financial situation. After extensive discussions with the local population, a flexible payment method was introduced, as it was considered a prerequisite for the acceptance of the whole project. Therefore, the option of Pay-as-you-go was favoured by all parties involved.

## 5. Discussion

The installation of the drinking water system results in a multi-layered socio-economic added value. Clean and safe water sources have a significant impact on the social context and are largely interpreted as a status symbol. Compared to neighbouring communities, this strengthens and activates the community and its cohesion. Furthermore, women and children are relieved, as they are predominantly responsible for the procurement of drinking water. For children, the extra time creates opportunities for educational activities, which increases their participation at school and results in better school-leaving qualifications, which reduces the risk of poverty. Women may use the extra time to earn additional income elsewhere.

In addition, the health of the affected people is directly influenced in a positive way. The availability of clean drinking water eliminates the risks of diarrhoeal diseases associated with contaminated water. Today, these diseases still lead to high child mortality. The risk of diseases such as schistosomiasis, hepatitis A, or trachoma is also significantly reduced. Overall, this not only increases life expectancy but also relieves households financially by eliminating possible treatment costs or costs of medication.

In Section 4.3, we calculated the effective cost price and operational and maintenance costs of the drinking water filtering system. The system can already pay for itself at a cost price of 0.70 EUR/$m^3$ in the course of the expected lifetime of 15 years. Compared to the three available alternatives for a safe and clean drinking water supply on Kibumba, which are bottled water (360 EUR/$m^3$), water from commercial tapping points (1.79 EUR/$m^3$), and water from neighbourhood taps (0.89 EUR/$m^3$), the cost price of 0.70 EUR/$m^3$ is 22% lower than the cheapest alternative (see Table 3). In addition, tapping points and bottled water are only available on the mainland, which involves additional effort and transport costs to Kibumba. Note that we did not take the expected inflation rate of 4.4% per year on average into account, as it does not only affect the maintenance costs of our system, it also adds to the sales price of the alternative approaches. In addition, it would also add to the sales price of our own overproduction. Instead, valid economic comparative data, especially inflation and wage development, should be recorded and evaluated in detail in further research work. Similarly, the electricity costs to operate the plant were not included

in our calculation as the local energy service provider (JUMEME) provides it free of charge with the support of the public sector to encourage further local development.

Overall, the initial situation for the project described in this article is very favourable, as the initial investments were made by external donors. Therefore, the revenue from the operation of the system can be used to cover the operating and maintenance costs only. In addition, financial reserves to replace the plant after its expected useful life of 15 years may already be built up during the operation period. However, although the investment costs of the plant were financed by third parties (10,640 EUR) the calculations in Section 4.3 show the economic viability of the proposed approach even if the financing costs are taken into account. This means that the annual operating and maintenance costs of EUR 1100 and the potential borrowing costs of 16.68% can be almost completely covered by the revenues from the plant at cost price.

In similar projects, the costs for drinking water treatment plants range between 2500–36,300 EUR [14] and 23,600–44,000 EUR, respectively. In addition, the annual maintenance costs of these alternative approaches range between 547–1921 EUR, which is also comparable to our approach (1100 EUR) [15]. Note that the operating and maintenance costs depend strongly on the technology and the scale of the respective plant. Therefore, the aforementioned ranges are relatively large. For example, some of the drinking water treatment plants do not require electricity for their operation. This means that communities without a mini-grid system may also operate the plant, which reduces its complexity as well as the maintenance and operational costs. However, the availability of electricity allows us to include an UV filtration stage, which increases the water quality significantly. In summary, this short comparison underlines that our approach lies in the lower cost range of comparable drinking water treatment systems. A detailed comparison between all different approaches is not within the scope of this article.

Even if the plant was not financed by the village community, the revenue model for the operation is important. For a suitable payment option, the living conditions on-site and the way people generate their income have to be considered. As almost all households on Kibumba do not have a fixed, regular income, a pay-as-you-go approach was chosen. The idea was to enable the highest possible flexibility for the local population and, thereby, increase the acceptance of the system.

The challenges of managing a self-sufficient drinking water supply are complex. Especially when individual requirements of the local people have to be taken into account. Therefore, actors from outside the community should not be responsible for the operation. This case study suggests a sustainable operation by the village community. The reasons for this approach are the following: First, ownership promotes careful and far-sighted management. Second, insufficient participation in public services, which also includes drinking water supply, is a major cause of displacement, especially in sub-Saharan Africa (SSA) [64].

In general, the success of these projects depends on the trust of the users. On the one hand, the water costs, and the way the system is operated determine the acceptance. On the other hand, the water quality has to be better or comparable to possible alternatives. In our case study, the effectiveness of the system is monitored in a daily routine check at the tapping point. Basic parameters such as the pH value, turbidity, colour, and coliform germs are measured via a quick test in addition to a visual inspection of the water quality. In order to further increase the acceptance of the plant, the village community should try to create jobs through the independent operation of the system, either via administrative or technical tasks. In this way, long-term effects could ensure the transformation of this approach into a sustainable socio-economic project.

During the installation, the islanders already had a very high interest in the plant as they are all affected by the health risks of unfiltered water. Another positive immediate association with the plant was the hope that they would not depend on expensive bottled water anymore, which would also relieve the island of the existing plastic waste problem. The fact that the approach is also of great political importance was made clear by both local

politicians and members of the national parliament who showed great interest directly on-site. There are already plans to develop a large-scale national program from this study.

## 6. Conclusions

In this article, we presented a case study about the installation of a drinking water filtering system on Kibumba island in Lake Victoria, Tanzania, Africa. Before the installation of the system, the local population covered their daily water supply either via bottled water, which was transported in plastic bottles from the mainland, or from the surface water of the lake. The installed drinking water filtering system has an annual capacity of 10,950 $m^3$ and was designed to cover the expected water demand on the island for the next 15 years. The system takes advantage of the availability of basic electricity, which was installed as a solar power mini-grid system on the island in 2019. In our economic analysis we calculated the water price for the sustainable operation of the system and compared it with existing water sources like neighbourhood taps, commercial tapping points, and bottled water from the mainland. It is demonstrated that the drinking water filtering system can be operated at a price of 0.70 EUR/$m^3$, which is approx. 22% more cost-efficient than the cheapest alternative (neighbouring taps). In our socio-economic analysis we worked out that the system is supposed to have positive effects in the social sector, it may improve the education of the local people and their health. Hence, this drinking water filtering system based on a photovoltaic mini-grid system may act as a role model to supply remote areas in countries with low access rates with clean drinking water.

**Author Contributions:** Conceptualization, J.F. and B.G.; methodology, J.F.; formal analysis, J.F., B.G and H.S.-M.; writing—original draft preparation, J.F.; writing—review and editing, J.F., B.G., M.A., W.S. and H.S.-M. All authors have read and agreed to the published version of the manuscript.

**Funding:** This research received no external funding.

**Institutional Review Board Statement:** Not applicable.

**Informed Consent Statement:** Not applicable.

**Data Availability Statement:** All data are included in the paper.

**Acknowledgments:** We would like to thank the reviewers for their valuable suggestions and comments to improve this paper. Financial support from a Rotary District Grant (C11413/2042/2199801) is gratefully acknowledged.

**Conflicts of Interest:** The authors declare no conflict of interest.

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
