# Peer review of "Drinking Water Supply in Rural Africa Based on a Mini-Grid Energy System—A Socio-Economic Case Study for Rural Development"

_sustainability, doi:10.3390/su14159458_

Round 1
Reviewer 1 Report
Dear colleagues,
Your manuscript, from its structure, is not really a scientific communication. The Introduction is very weak. It is only in the results I have realized that it is not just a concept but a real case study.
I have attached my lecture notes. The (approach and the) results are excellent. You should no present your results in the spirit in them you acquired them, but rather in the perspective to add knowledge.
Reviewer X

Author Response
The authors would like to express their sincere gratitude for the time and effort invested in reviewing the manuscript, as well as for the helpful comments and appreciation of the submitted work. We appreciate the many constructive comments and hints, which helped us a lot to improve our manuscript significantly.
Please find attached our response to the comments.
With kind regards,
Joern Falk

Reviewer 2 Report
-
This study is very relevant given that the global water supply is decreasing and there is a need for new and innovative sources of fresh water.
-
There are many sentences where a citation is required but not present. Please review the introduction carefully and add in citations where appropriate and necessary.
-
Please remove the capitalization of “FIGURE” and “TABLE” throughout the document.
- Please see the attached PDF for additional comments.

Author Response

(The authors gave the same response as above.)

Reviewer 3 Report
This is a reasonably well-written but somewhat long-winded paper that, although it seems to have little novelty, provides an interesting case study of the economics of a small decentralized drinking water supply system in Tanzania.
The paper provides a clear overview of the local situation and relevant considerations. The economic consideration of the project seems incomplete, though:
- The energy supply system is treated as a given, with no costs for its use, maintenance or eventual replacement included in the costs for the water supply system.
- These same is true for a distribution infrastructure. Are people going to carry away 12 or 30 m3 per day in jerrycans? I have no idea.
- There is no consideration of inflation in the projected costs for the coming 15 years. This does not seem realistic.
Some additional remarks:
- I think that the paper can be shortened significantly without losing any relevant information.
- There is no reference to other comparable projects in developing countries, though there must be many that have been described in the literature.
In addition to this, I have a number of more minor comments, see below. I recommend that these be addressed before possible publication.
There is significant overlap between the introduction and section 2.2 I think that the former can be shortened, moving parts of it to the latter.
Several occurrences of “. (CITATION).”; please remove the first period.
Several occurrences of TSZ (Tanzanian Shilling) which should be TZS.
14: perhaps growing food and personal hygiene outrank recreational activities.
66: Lake Victoria is the world’s second-largest in terms of surface area, not of volume.
74-75: Might agriculture also be an important source of nitrates? And natural sources for fluorine?
109: schistosomes are neither bacteria nor viruses, but flatworms; suggest to remove “other”
163-164: What is the expected remaining life of these batteries?
172: presumably kWh/m2/yr?
177: insert “of the”, ON -> on, ISLAND -> Island
221: annually
265: Correct capitalization and remove italics. Remove horizontal line above the last row.
282-283: FOR -> for, drinkung -> drinking, Based -> based
Table 2: no acceptance criteria for smell? Also, I am unfamiliar with the use of the word “scope” in this context.
296: The water is taken from the lake?
297: and other pathogens?
327, 330, 341: Correct capitalization.
348: Mwanza
Table 3, line 432: Please use the same exchange rate for all prices in EUR.
361-362: Correct capitalization.
385: That is yet another exchange raet from the two used in Table 3; please be consistent.
387: Szeanario?
Table 5: The annual maintenance costs are surely not going to remain the same over 15 years. How much is inflation in Tanzania on average?
Table 6: How is this 30 m3/day going to be transported to its users? Does that not also require an infrastructure with its own capital and operational costs? These have not been included in Tables 4 and 5.
Table 6 #4 and line 408: This should be the annual costs, i.e. 1100+1970=3070 EUR, because you are aiming to recover these (“price per m3 at cost”). 3066 presumably includes a rounding error.
405: Please use “,” to separate thousands.
503: First mention of “bilharzia”, please also mention when discussing schistosomes (line 109).
658, 701: No full capitalization of authors required.
Author Response

(The authors gave the same response as above.)

Round 2
Reviewer 1 Report
Dear colleagues,
Congratulations for the successful revisions. THere are still some small aspects to address including ordering the key words. There are also section where small elabporations are very useful. Just go through the manuscript (example below from the introduction).
A weakness I had not noticed before is the TOO high ptoportion of technical documents in the references. For future articles you should consider that aspect; Hence, alternative approaches for connecting disadvantaged areas have to be investigated (refs). Many interesting approaches have been developed in recent years (refs. with elaboration. Shannon et al. 2008 (at Nature) is one of the best references here). The focus was on robust and autonomous technologies for water treatment systems [9]...
Sincerely,
Reviewer 2
Author Response
Dear Reviewer,
The authors would like to thank you once again for the time and effort you have invested in reviewing the manuscript. Your helpful comments and constructive advice have helped us to improve our work once again.
Kind regards

Reviewer 2 Report
Thanks to the authors for making the requested changes to the manuscript. There are still additional changes that can be made for further improvement before publication. Please make the changes suggested in the comments of the attached PDF. Additionally, please carefully review the paper for areas where sentence structure and grammar can be improved. There are several sentences that would benefit from more finesse in the authors' writing. When resubmitting, please submit both a marked up copy and a clean copy without any tracked changes. Thank you.

Author Response

(The authors gave the same response as above.)
